# Spotlight on New Hallmarks of Drug-Resistance towards Personalized Care for Epithelial Ovarian Cancer

**DOI:** 10.3390/cells13070611

**Published:** 2024-03-31

**Authors:** Simona Frezzini, Sara Lonardi

**Affiliations:** Unit of Medical Oncology 3, Veneto Institute of Oncology IOV—IRCCS, 35128 Padova, Italy; sara.lonardi@iov.veneto.it

**Keywords:** epithelial ovarian cancer, drug-resistance, biomarkers

## Abstract

Epithelial ovarian cancer (EOC) is the deadliest gynecological malignancy worldwide. Despite the latest advances, a major clinical issue in EOC is the disappointing prognosis related to chemoresistance in almost one-third of cases. Drug resistance relies on heterogeneous cancer stem cells (CSCs), endowed with tumor-initiating potential, leading to relapse. No biomarkers of chemoresistance have been validated yet. Recently, major signaling pathways, micro ribonucleic acids (miRNAs), and circulating tumor cells (CTCs) have been advocated as putative biomarkers and potential therapeutic targets for drug resistance. However, further investigation is mandatory before their routine implementation. In accordance with the increasing rate of therapeutic efforts in EOC, the need for biomarker-driven personalized therapies is growing. This review aims to discuss the emerging hallmarks of drug resistance with an in-depth insight into the underlying molecular mechanisms lacking so far. Finally, a glimpse of novel therapeutic avenues and future challenges will be provided.

## 1. Introduction

EOC is the deadliest gynecological malignancy and an extremely heterogeneous disease [1,2,3]. The gold standard treatment of newly diagnosed EOC includes optimal cytoreduction, namely primary or interval debulking surgery, along with platinum chemotherapy backbone, with carboplatin-paclitaxel regimen being the therapeutic mainstay frontline [2,3]. In the last decade, the advent of strategies in the maintenance setting targeting tumor angiogenesis and DNA repair, such as bevacizumab and PARP inhibitors (PARPis), has ushered in a new era of anticancer therapy for EOC [4]. PARPis are new standard-of-care maintenance regimens licensed both in frontline and platinum-sensitive relapse settings, regardless of BRCA and homologous recombination deficiency (HRD) mutation status. Olaparib plus bevacizumab maintenance has been licensed as a frontline for HRD-positive patients irrespective of their clinical risk [4]. Despite maximal surgical and therapeutic efforts, EOC treatment remains a big challenge due to late-stage diagnosis, the high rate of early relapse despite the initial platinum response, and the dismal prognosis related to upcoming chemoresistance [3,4,5]. Clinical heterogeneity appeared in terms of disease staging, progression kinetics, and treatment response. An intriguing trait of recurrent EOC is the successful platinum rechallenge, mainly carboplatin with partner drugs, such as paclitaxel, pegylated liposomal doxorubicin (PLD), and gemcitabine, leading to a sustained remission accompanied, as a rule, by progressive shortening of the intervals between relapses until platinum sensitivity is lost. PFI (platinum-free interval), causing stratification into platinum-refractory (PFI < 1 month), partially platinum-sensitive (PFI 6–12 months), and platinum-sensitive (PFI > 12 months) relapse, has evolved into the treatment-free interval (TFI) concept [6]. Limited knowledge of chemoresistance mechanisms makes the treatment of relapse a dilemma. EOC can be regarded as a suitable model for exploring the key role of CSCs as primers for disease relapse/progression [7,8]. CSC-driven repopulation is a prominent phenomenon in the clinical course of this disease; therefore, it is a turbulent research field [8].

No predictive markers of primary resistance to platinum/PARPis at diagnosis have been validated yet [2]. Due to the complexity of the pathways involved in EOC progression or recurrence, reliable biomarkers are eagerly awaited to identify treatment refractory EOC [6]. The prognostic role of Cancer Antigen 125 (CA-125) in EOC is widely acknowledged [2]. Retrospectively, early level changes in CA-125 up to the third chemotherapy cycle showed prognostic significance in terms of OS and PFS for advanced EOC patients [9]. Although being routinely used for evaluating disease status and monitoring treatment efficacy [2,9], there is less certainty about its use for follow-up [2].

Despite several attempts to characterize the molecular features of chemoresistance, a rift between promising preclinical findings and unsuccessful clinical translation to patient survival still exists [10,11]. Recently, crosstalk among major tumor cell-intrinsic and extrinsic signaling pathways has been argued to drive tumorigenesis, epithelial-to-mesenchymal transition (EMT), and drug resistance [12,13]. Moreover, the identification of candidate miRNAs involved in stemness and chemoresistance may add to the therapeutic potential of miRNA dysregulation in EOC [13]. Hypothesis-generating data supporting the role of CTCs in drug resistance provides a rationale for validating them in large multicenter trials as a tool for treatment stratification [14]. Lastly CTCs, miRNAs, and other surrogate biomarkers of drug resistance mandate prospective investigation and extensive clinical validation before routine implementation [12,15,16,17]. 

In the rapidly evolving therapeutic landscape of EOC, early diagnosis is unsatisfactory and platinum resistance is a highly relevant issue. The successful management of disease relapse and resistance is an unmet clinical need. The development of reliable biomarkers serving as clues for personalized care is a current endeavor. The present narrative review aims to provide in-depth insight into emerging hallmarks and putative biomarkers of multilayer heterogeneity and drug resistance in EOC [4,12,13]. An overview of EOC complexity (Figure 1) may enable successful clinical translation into effective therapeutics [10] towards the prioritization of personalized therapy for resistant diseases.

## 2. Materials and Methods

The Literature searches for the present narrative review were initially conducted in PubMed for papers published up to January 2023, using the following search terms: (chemoresistance OR drug resistance OR heterogeneity OR biomarkers) AND (recurrent epithelial ovarian cancer). A PubMed (PubMed, RRID:SCR_004846) search alert was used to capture additional articles published between March 2023 and July 2023. Searches were restricted to ‘drug resistant epithelial ovarian cancer’, ‘biomarkers’, cancer stem cells, and targeted therapies. The 40 articles retrieved from the above sources included preclinical studies, in vitro studies, early phase clinical trials, and review articles providing a rationale for chemoresistance and potential targeting therapeutics. They also included Phase II–IV clinical trials, leading to the approval of established standard strategies and key trials conducted thereafter. Thus, key papers were included based on the authors’ clinical experience and knowledge of the field.

## 3. Results

### 3.1. What Is the Roadmap of Multilayered Heterogeneity in EOC?

Clinically, histologically, and molecularly, EOC is a highly heterogeneous disease. The overall heterogeneity of EOC observed in clinical settings could be deemed as the outcome of distinct and interlinked layers of EOC complexity along the entire tumor evolution path, directly and indirectly involving the CSCs subset [8] (Figure 1).

#### 3.1.1. Clinical and Histopathological Heterogeneity of EOC across the Main Subtypes

Knowingly, OC represents an umbrella term for various subtypes of malignant diseases, all involving the ovary but not necessarily interrelated to it and sharing key clinical features, especially the pattern of metastatic dissemination. Unlike other neoplasms, EOC metastasis is largely limited to the peritoneal cavity and surrounding organs, with only a minor role in hematogenous spread [8]. EOCs, also designated as tubo-ovarian malignancies [2], constitute 90% of all OCs and can be further grouped into five major histopathological subtypes, as follows: high-grade serous ovarian cancer (HGSOC), 70%; low-grade serous ovarian cancer (LGSOC), <5%; endometrioid ovarian cancer (ENOC), 10%; clear-cell ovarian cancer (CCOC), 6–10%; and mucinous ovarian cancer (MOC), 3–5% [15,16]. They differ in terms of origin, prognosis, biology, and clinical and molecular profiles [16]. Moreover, a novel two-tier classification system that combines clinicopathological data with molecular features designates EOC as low-grade type I, including LGSOC, MOC, ENOC, and CCOC, and high-grade type II, including HGSOC, carcinosarcoma, and undifferentiated carcinoma [8]. Type I tumors develop from implants in the ovary of benign extraovarian lesions and then switch to a malignant genotype/phenotype, while type II tumors arise from serous tubal intraepithelial carcinoma/neoplasia (STIC/STIN). From a biological standpoint, type I neoplasms show a more indolent behavior and genetic stability, as compared to type II neoplasms, which exhibit a higher disease burden along with widespread genomic and chromosomal instability [7]. Based on expression profiling studies, most of the recurrent mutations encountered in non-HGOCs affect mitogenic signaling converged at the MAPK pathway [8], promoting survival and chemoresistance [6]. In more detail, the most common aberrant pathways in type I neoplasms are BRAF, KRAS, WNT-ß-catenin, PTEN-PI3K, ERBB2, and ARID1A, as compared to those found in type II neoplasms, such as p53, RB1, NOTCH3, AKT, BRCA1/2, HER-2/HER-3 overexpression, and p16 inactivation [7,15]. 

HGSOC accounts for 70–80% of EOCs and represents the highest disease burden and mortality in EOC due to its more aggressive behavior, later-stage diagnosis, and earlier relapse despite optimal chemosensitivity. Hence, EOC is often referred to as “silent killer” or “whispering disease” [8]. The biological dichotomy among EOC subtypes, leading to the two-tier binary system, reflects the higher mutational load of HGS compared to the limited mutational signature of its counterpart [8]. Accordingly, the type II class is intrinsically heterogeneous, with further definable molecular subtypes. The Cancer Genome Atlas (TCGA) project [17] separates HGSOC molecular subtyping into four distinct phenotypes, C1/mesenchymal, C2/immune-reactive, C4/differentiated, and C5/proliferative, featuring diverse clinical behavior due to their interplay with stromal cells in the tumor microenvironment (TME) [8,10,18]. Poor prognosis has been reported for both C1 and C5 subtypes [6,10].

#### 3.1.2. Developmental Heterogeneity of EOC

EOC is retained as a stem cell (SC)-driven tumor type, although the exact cell of origin remains elusive, to date [8]. Irrespective of the cell-of-origin, the EOC itself and its SC component feature intrinsic heterogeneity [8]. The periodic replacement of ovarian surface epithelial (OSE) cells lost during ovulation strongly suggests the need for a long-term proliferative reserve arising from self-renewing SCs, also known as OSE SCs [8,19]. The existence of such SCs and the anatomical location with OSE have been recently debated. Long-term stem cell maintenance is ensured by tight modulation of Wnt/β-catenin signaling [8]. Reportedly, even the fallopian-tube (FT) epithelium, albeit being more mature than the OSE, contains putative secretory SCs capable of self-renewal and progenitor cell differentiation through Wnt/β-catenin signaling [8]. The critical cellular source providing the Wnt ligands to the OSE and FT SCs is still enigmatic [8]. Notably, both the OSE SCs and FT SCs rely on the same molecular pathways and overlapping niche requirements [8]. 

Historically, OSE SCs have been regarded as the prime candidate for ovarian carcinogenesis [20] based on the “incessant ovulation hypothesis”, leading to genomic defect accumulation and likely malignant transformation. Histologically, the lack of precancerous changes in the ovary, both in advanced disease and in prophylactic oophorectomy from healthy BRCA-mutated patients, opposed to early dysplastic lesions found in the FT epithelium in the same setting, posed a diagnostic challenge to the pathologists. Moreover STIN/STIC [8], harboring p53 and PAX8 markers on immunohistochemical (IHC) staining, was found in proximity to the tubal–peritoneal junction [8,19]. All these findings led to the conceptual framework of the FT epithelium as the candidate source for EOC (especially HGSOC) development [8]. This theory is deemed to be complementary, rather than contrasting, to that of incessant ovulation, as the ovulation-related wounds may favor the entry of migratory FT cells with tumor-initiating capacity [8]. Thus, the suggested developmental origin of type I non-HGS tumors is variable (OSE, endometrium, or extra-Müllerian tissue) as compared to that of type II HGS tumors (FT) [8].

Accordingly, the ovary is deemed a fruitful niche for dispersed cancer cells, as well as a first site for metastatic spread, suggesting its loco-regional involvement through an adjacent, female-specific tissue. EOC heterogeneity may rely on phenotypic variations and the early niche of the cell of origin, contributing to a broad range of clinical presentations [8]. 

#### 3.1.3. Cellular Heterogeneity of EOC

CSCs can now be retained as the milestone of drug-resistance in EOC. EOC development and chemoresistance may rely on a combination of both clonal evolution and hierarchical CSC models [7,8]. EOC heterogeneity is a byproduct of these models and is clinically modified by treatment regimens. The CSC model relies on the expansion of pre-existing “stem-like” precursors harboring constitutive resistance to cytotoxic/cytostatic agents because of continuous Darwinian selection under treatment, without significant changes in the overall tumor mutational load. The resultant tumor cells show a hierarchical inheritance pattern from their initiating CSCs, along with unique phenotypes owing to the diverse mutations and activation pathways acquired during differentiation [7]. Ovarian CSCs are thought to represent only 1% of the EOC cell repertoire [10]. They exhibit stem-like properties, such as tumorigenicity in vivo, asymmetric cell division [8], and invasiveness because of further genetic or epigenetic defects [21]. Unlike normal SCs, they show no homeostatic balance between self-renewal and pluripotency, thus promoting malignant transformation. Like their counterparts, CSCs show differentiation and DNA repair capabilities, as well as multidrug resistance (MDR), by altering drug transporters [22]. The maintenance relies on their highly specialized microenvironment, namely niche, which is made up of extracellular matrix (ECM) and stromal cells and is anatomically distinct within the overall TME. Crosstalk between CSCs and their niches promotes genotypic/phenotypic diversification [22] and cancer stemness, the molecular basis of which is the network among the major signaling pathways [23]. Recently, a new statement on cell plasticity, referring to the bidirectional switch between stem and non-stem-like phenotypes, posed additional challenges in this field. Notably, CSCs escape conventional chemotherapy owing to their quiescent nature, thus representing a major source of chemoresistant cells within tumors [21]. Clinical analyses of matched primary/recurrent EOC samples revealed CSCs pool enrichment during chemotherapy, pursuing self-renewing effectors of chemoresistance [21]. Outgrowth of residual CSCs in their niche after primary therapy prompts disease relapse [6]. CSC plasticity and dormancy reliably underlie the occurrence of local/distant relapse after long delays and therapeutic resistance [21]. 

At the cellular level, the main platinum-resistance mechanisms of CSCs, which vary among histotypes, are (a) enhanced platinum efflux pumps (ATP binding cassette family), ensuring genome integrity defense against chemotherapeutics; (b) increased sequestering/inactivation of platinum and repair of platinum-induced DNA damage; (c) decreased platinum uptake; and (d) increased anti-apoptotic signaling [21]. Primary “platinum refractory” (mostly non-HGS) EOCs are intrinsically drug resistant with very early relapse/progression during/after treatment. Moreover, in tumors showing an initial platinum response, the equilibrium between sensitive and resistant subsets establishes the final tumor response to the platinum backbone [21]. Platinum resistance is a therapy-oriented definition of EOC. The currently preferred regimens for chemoresistant disease include sequential use of non-platinum drugs (as reported in Appendix A). In this regard, residual toxicity from prior therapies, drug accessibility, and patient–clinician agreement may inform the treatment choice [2,3,6].

The CSC phenotype relies on enhanced expression of putative surface and intracellular markers, reinforcing the significance to chemoresistance (as detailed in Appendix A) [8,24]. Recently, some putative markers like CD44, CD24, CD133, SOX2, and aldehyde dehydrogenase (ALDH) have been proposed but with vaguely defined phenotypic features due to the consistent phenotypic and functional plasticity of CSCs [8,22]. Of note, only a combination of these markers may help detect CSCs compared to single markers. The enhanced expression of CSC markers at both protein and mRNA levels post-platinum chemotherapy reinforces the significance of CSCs to chemoresistance as well as their role as biomarkers for EOC progression [22]. However, CSCs display ambiguous phenotypes owing to marked phenotypic and functional inter-/intratumoral heterogeneity [19,20]. Intratumoral heterogeneity is both spatial within the primary tumor and temporal between the primary tumor and its metachronous metastases, as shown in biopsy samples performed at different time points during the clinical journey. Hence, there are clinical and radiological findings of differential treatment outcomes (progression and responses) within the same tumor [21]. Accordingly, CSC phenotypic plasticity may be the main mechanism for long-term treatment failure in EOC and spontaneous escape variants due to minimal residual disease (MRD) [8]. Despite the ambiguous phenotype of ovarian CSCs, CSC-specific markers, especially in combination, may act as valuable platforms for EOC biomarker discovery [22]. CSC heterogeneity is further corroborated by intratumoral variability within the CSC compartment due to additional genomic or epigenetic changes [8] without impairing biomarker inference from the transcriptomic landscape [8].

Another topic of interest in this field is the interplay among EMT, CSCs, and chemoresistance. Owing to the CSC dynamic state, “stemness” induction is primarily due to exogenous factors within the CSC niche. The underlying epigenetic event is EMT, a well-known mechanism of platinum resistance, which enables epithelial to mesenchymal cell differentiation for EOC progression or metastasis. It is marked by morphological changes, decreased cell–cell adhesion, loss of cell polarity, gain of cell motility, ECM remodeling, and gene expression patterns crucial to metastatic spread and chemoresistance [10]. The key pathways driving EMT are TGF-β, PI3K/AKT/mTOR signaling, and MDR [24]. The shift towards a mesenchymal state also provides synergism between CSC markers and EMT-related factors, enhancing CSC evolution and, ultimately, chemoresistance. The EMT gene signature acts as a negative prognostic factor in HGSOC, indeed. Future studies exploring the interplay between CSCs and EMT are needed to deepen the understanding of EOC relapse [24]. 

Recently, the major signaling pathways actively involving CSCs have been highlighted. Notably, the most pertinent pathways to advanced EOC are directly or indirectly involved in the maintenance, self-renewal, and drug resistance properties of CSCs (Appendix A) [22,25]. 

Dysregulation of the major signaling cascades, such as MAPK, PI3K/PTEN/AKT, JAK/STAT3, Notch, and NF-KB, promotes malignant tumor phenotypes (chemoresistant, metastatic, and proliferative), ultimately leading to poor clinical outcomes [25]. Nonetheless, their implication in normal homeostasis makes therapeutic targeting challenging [7]. Crosstalk is likely between them. Therefore, a deeper insight into these main stemness effectors and their networks would provide a platform for identifying viable therapeutic targets [7,22].

#### 3.1.4. Microenvironmental Heterogeneity in EOC

A growing body of evidence supports the vital role of non-cancer cells in the TME in tumorigenesis, EMT, metastasis, and drug resistance [10]. The TME molecular milieu, as shown by -omic tools, consists of the ECM and stromal and immune cells [11]. The diversification of TME through dynamic molecular events is a hallmark of EOC heterogeneity [11]. The TME is also involved in tumor-related metabolic reprogramming [26]. Unlike their counterparts, chemoresistant CSCs are reliant on oxidative phosphorylation (OXPHOS)-mediated lipid metabolism. The metabolic heterogeneity or plasticity of CSCs allows for the switching of metabolic flux from OXPHOS to glycolysis [26]. The flexible metabolism of CSCs is due to the TME, therapy-induced changes, and nutritional requirements [26]. An attractive strategy could be the simultaneous targeting of all metabolic compartments in the TME to boost cancer control and eradicate chemoresistant CSCs [26].

The reliance of CSCs on their niches is a current area of research [24]. The dynamic state of CSCs is influenced by the TME [10]. In turn, CSCs shift from a pro-inflammatory to a pro-tumorigenic immune system [22]. Recently, the co-existence of both progressing and regressing metastases, with immunosuppressive and immune-activating patterns, respectively, in the same tumor of the same patient has been argued. Novel TME-targeting approaches have been tested to reverse chemoresistance [10,11]. Among them, only anti-VEGF bevacizumab has been standardized and incorporated into current guidelines, both as a frontline therapy, being the preferred option in BRCA-negative patients with high-risk disease, and as a gold standard in platinum-sensitive relapse [4]; it can also be used as a non-standard option in platinum-resistant EOC [4]. 

To the best of our knowledge, there are no prospectively validated biomarkers for the response to bevacizumab. The only significant clinical predictors of the benefit of bevacizumab in frontline therapy were performance status, stage, and residual disease after primary surgery [27]. To date, decision making regarding maintenance therapy relies on patient-related factors, such as platinum sensitivity, BRCA and HRD status, disease burden, and expected toxicities [28]. Collectively, the TME is emerging as a new attraction to target drug-resistance in EOC. Nonetheless, further investigation is mandated. 

#### 3.1.5. Heterogeneity of Molecular Milieu in EOC

Highlights from TCGA elucidate the processes underlying drug resistance in EOC [10]. Molecular genetic and epigenetic aberrations vary by histotype. Genomic instability is a phenotypic hallmark of HGSOC, mainly due to ubiquitous TP53 mutations and BRCA 1/2 defects [19]. The intratumoral heterogeneity of HGSOC reflects the clonal evolution that occurs during tumor progression [19]. Significant spatial heterogeneity occurs first at the level of the mutational profile between the primary tumor and peritoneal metastases, leading to multiple populations of genetically and phenotypically distinct subclones evolving from an ancestral clone [19]. Subclonal tumor cell proliferation also promotes a variable mutational landscape within the primary tumor and its metastasis that, even through the platinum chemotherapy’s selective pressure, fosters the temporal diversification of the recurrent tumor mutational landscape from the original one [29].

Considering the TCGA project, homologous recombination repair (HRR) deficiency was found as a hallmark and a crucial therapeutic target of HGSOC [29]. The HRR system, the critical eukaryotic pathway enabling high-fidelity repair of double-stranded DNA (dsDNA) breaks, relies on several proteins including BRCA1-2. Deficiency in DNA damage repair due to dysfunctional HRR is also referred to as a HRD signature, which is broadly identified in about half of HGSOC patients [18]. The HRD-related genomic instability has gained supremacy in the context of the HGSOC mutational landscape. Germline or somatic HRR deleterious mutations, firstly affecting BRCA 1-2 tumor suppressor genes, are detected in up to 30% of HGSOC cases, mainly accounting for hereditary EOC [30]. Further genetic/epigenetic defects included in the HRD signature identify somatic mutational landscapes reflecting the BRCA-like or BRCAness phenotype. The HRD signature may serve as a biomarker for platinum and PARPi sensitivity [31] and thus may inform prognosis and treatment decision making of BRCA-like tumors, translating into longer survival time and time to platinum resistance [30]. 

Currently, BRCA 1/2 testing is recommended for all patients diagnosed with EOC, FT, or primary-peritoneal (PP) cancers irrespective of family history, with both tumor and genetic testing performed only in tBRCAm-carrier patients [4]. HRD testing at primary diagnosis can broadly identify HGSOC patients who are most likely to benefit from PARPi. However, further prospective data are needed to recommend tumor testing for non-BRCA HRR mutations due to mostly negative results regarding their predictive value [30]. Strikingly, in a platinum-sensitive relapse setting, the benefit of HRD testing is impactful to a lesser extent. Historically, the milestone of the PARPi mechanism of action is known as “synthetic lethality”, which consists of a loss-of-function mutation of BRCA genes coupled with synthetically inhibiting PARP1. The stalled replication forks due to PARP1 blocking enable genomic instability and cell death [30]. 

Chemoresistance can rely on both primary and acquired (restored) HRR proficiency either at the first-line or later lines [6,31]. Emerging hallmarks of acquired drug-resistance have been highlighted, to date. In clinical and preclinical tBRCAm tumors, secondary somatic reversion mutations (e.g., missense, splice reverted variants, and deletion/insertion) rescued BRCA function as well as functional HRR in mBRCA1-2 HGSOCs throughout the treatment course [31]. Additional mechanisms of acquired resistance under platinum or PARPi selection pressure, also termed “somatic plasticity”, have been reported, for instance, the re-start of replication forks. Intriguingly, in EOC cells from PARP inhibitor-resistant patients, the enhanced HRR proficiency was reported to be due to a survival advantage of CSCs over PARPi synthetic lethality [6]. The current challenge is to avoid disease recurrence in BRCAness tumors. Collectively, all emerging mechanisms of platinum/PARPi resistance in BRCAness EOCs could be further exploited to validate biomarkers that are critical for the early detection of reversions and outcome prediction [31]. 

Recently, epigenetics has been proposed to explain the main mechanisms underlying drug resistance in EOC. The drug-adaptability or plasticity of CSCs, consisting of rapid reversion of drug-resistant CSCs into drug-sensitive subsets, is not driven by heritable gene mutations but by a poised epigenetic state [22]. Histone modifications favoring a chromatin bivalent state are directly implicated in the epigenetic regulation of CSCs and their acquired drug resistance (as summarized in Appendix A) [22]. Histone deacetylation and DNA methylation are players in epigenetic silencing, accounting for drug resistance; therefore, their reversion mechanisms are likely to restore drug sensitivity in cisplatin-resistant cells [13]. Hence, epigenetic changes serve as possible targets for eradicating drug-resistant populations and overcoming the reversible resistance of CSCs [7,22]. 

### 3.2. Any Viable Options and Challenges in Targeting Drug Resistant CSCs?

One of the major challenges in the successful treatment of EOC is the development of recurrent and chemoresistant disease. It is noteworthy that the transcriptomic profiling in the TCGA project was bulk RNA sequencing rather than single-cell sequencing, thus overtaking the role of individual cell types in intratumor heterogeneity within the TME [14]. Notably, the paucity of secondary debulking or biopsy samples limited the studies testing CSCs on chemoresistant tissues compared to their chemo-naïve counterparts. Focused biopsies after treatment may help define dynamic changes due to the TME [22]. 

The CSC dynamic state hampers CSC identification and sorting [22] as well as the design of straightforward therapeutic targeting [8,10] due to the lack of a universal chemoresistance signature. In this regard, targeting stromal cell microniches in the TME and/or their EMT-related factors is thought to be a more successful strategy given their role in stemness induction [11]. For instance, stromal cells were preliminarily identified, but the main population providing niche activity for CSCs via paracrine/juxtacrine factors needs to be clarified [8]. 

The discovery of highly tumorigenic and chemo-resistant CSC-specific biomarkers paves the way to development of CSC-targeting therapies to counteract tumor relapse and resistance. At present, the primary effort has been the identification of prognostic biomarkers [19] but falls short of informing treatment selection [10]. In this regard, targeting TME stromal cells and/or their factors pursuing EMT may serve as a successful strategy since they are known to influence stemness [10].

### 3.3. Are There Any Candidate Biomarkers for Prognosis and Outcome Prediction?

#### 3.3.1. miRNA, Exosomes, and Chemoresistance

A growing body of work has focused on the biology of miRNAs and their roles in modulating EOC chemotherapeutic sensitivity [13]. To date, 39 miRNAs have been found to be aberrantly expressed in EOC through genomic or epigenetic mechanisms, with highly relevant roles for miR-200 and Let-7 families [19]. Due to their roles in EMT, CSCs, and cancer metastasis, miRNAs act as hopeful diagnostic and prognostic biomarkers as well as potential therapeutic targets in the future [32]. However, their exact role in chemoresistance remains to be clarified. The blockade of EMT may help overcome drug resistance, as well [32]. Most EMT-related miRNAs, such as miR-200 family members, are markedly downregulated in EOC tissues during the EMT process, thus suppressing metastasis spread through the EMT regulation [32]. An in-depth insight into many miRNA functions and their interplay with the P13K/AKT/mTOR pathway has revealed both oncogenic and tumor suppressor miRNAs, thus advancing the identification of surrogate biomarkers of drug sensitivity [13] and potential miRNA-targeting agents. Overall, oncogenic miRNAs are thought to promote EMT, chemoresistance, and poor prognosis [13], in contrast to tumor suppressor miRNAs that restore platinum sensitivity in OC cells [32]. Notably, highly stable miRNAs enriched in exosomes undergo paracrine trafficking from cancer to non-cancer cells to induce chemoresistance. Exosomal miRNAs are critical mediators of the crosstalk between tumor and stromal cells [19,32], providing pre-metastatic niche preparation by a disrupted regional vascular supply and an immunosuppressive TME towards metastasis [32]. Manipulation of this milieu may provide future therapeutic targets [15]. 

#### 3.3.2. Liquid Biopsy-Based Biomarkers in EOC

Liquid biopsy-based noninvasive biomarkers, mainly cell-free DNA (cfDNA) and CTCs, are being addressed in EOC with an emerging role in disease management [16,17]. Their main advantage is the real-time monitoring of disease status by the detection of tumor-specific changes in cancer progression, which is amenable to capture tumor heterogeneity. Thus, cfDNA genomic and epigenomic profiles can be highly dynamic between primary and metastatic tumors and within a single patient at different time points. However, standardization of the analysis platforms for these biomarkers is needed to ensure their reproducibility. Combinatory approaches of CTCs and cfDNAs may maximize liquid biopsy efforts towards the individualization of anticancer therapy in EOC [16]. 

First, cfDNA analysis in EOC is thought to be promising for diagnosis, prognosis, monitoring of MRD, platinum/PARPi resistance evolution, treatment response, and decision making. Next-generation sequencing (NGS)-based methods are employed for downstream analysis of cfDNA and provide comprehensive mutational profiling in EOC. Molecular characterization of cfDNA is a valuable tool for personalizing EOC anticancer therapy [16]. 

CTC research is limited in EOC, likely due to their rarity and heterogeneity, preventing successful detection, while their predictive value needs to be clarified in this setting. CTC clusters have been reported to display special drug-resistant phenotypes [12,14,15]. CTC tests are not routinely incorporated into the current guidelines [16,17]. Nonetheless, the latest advances in the automation of CTC platforms are paving the way for their clinical applications. In fact, CTC tests may serve as a “real-time liquid biopsy” to predict drug resistance and detect MRD after optimal debulking [12], thus leading to risk stratification of adjuvant treatment for patients [12,14]. Unlike “proof-of-concept” data [12,13,16], a prospective comparative analysis of CTCs in the chemoresistant/chemosensitive populations showed no significant predictive value for chemoresistance. Hopefully, progress in the -omic profiling of CTCs and detection methods may help identify correlations with chemoresistance [12]. 

## 4. Discussion and Future Challenges

The multilayered complexity of EOC is likely to counteract the long-term success of cancer therapies [8]. Reportedly, crosstalk between CSC markers and CSC niches paved the way for directed therapeutic target design [7]. CSC-targeting strategies remain a challenge until in vivo or in vitro surrogate assays to identify CSCs are developed [33]. Proof-of-concept data may reveal novel targets in the TME to be harnessed for CSC-directed therapies, ultimately leading to a paradigm shift in EOC therapy [24] (Figure 2). Better insight into the detailed landscape of the TME may trigger the deployment of molecularly informed treatments for drug resistant EOC [28]. 

Current research is focused on multiagent therapies involving PARPi, cytotoxics, antiangiogenics, and multikinase inhibitors [6,25,31] and is expected to successfully reverse chemoresistance and improve long-term patient prognosis [10]. Of note, promising therapeutics for the chemoresistance pathways are expected to emerge in the future based on innovative studies (Appendix A). The notion of TME significance in EOC chemoresistance has ushered in a new appreciation for EOC complexity and likely targeted strategies (Appendix A) [10]. A growing body of work has argued the therapeutic effect of immune checkpoint inhibitors (ICIs) as a potential route to restore the antitumor immune response by TME modulation. Accordingly, a new era of combinatorial immunotherapy holds promise for refractory EOC treatment [25,35]. 

Furthermore, epigenetics inhibitors are being tested to restore platinum sensitivity, but none of them have been licensed for EOC [6]. Recently, preclinical and early phase clinical trials have focused on the PI3K/AKT/mTOR, Wnt, Notch, Hh, and YAP/TEAD signaling pathways, through a broad range of targeting approaches, in an attempt to restore EOC chemosensitivity [7,36]. Disappointingly, no pathway-directed inhibitors have progressed to late-phase clinical trials [6]. Early encouraging data for Hh inhibitors hint at their ability to influence CSCs [7]; however, further research hypotheses regarding these signaling pathways need to be elucidated [7,21]. Exosomes harboring oncogenic promoters may serve as potential drug delivery systems, providing disease stability and antitumor immune responses [25]. 

Recently, the efficacy and clinical utility of biomarker-driven targeted therapy have been suggested, warranting further exploration in phase 3 trials (details are listed in Appendix A) [37,38,39,40]. With recent advances in -omics tools, it is the right time for this endeavor. A deeper outlook on the evolving mechanisms of platinum and PARPi resistance in BRCAness EOC will pave the way for novel targeting options, such as combined ICI or antiangiogenics with the current standard of care [31]. Translational research including -omic-based tools should be incorporated into prospective trial designs to help uncover new hallmarks of drug resistance, validate putative biomarkers, and launch reliable therapeutics for relapsed/resistant diseases [6].

## 5. Concluding Remarks

Presently, the discovery of molecular predictors paving the way to targeted therapeutics is a current effort in EOC. Patient risk-tailored treatment stratification and biomarker-driven strategies are needed to hopefully overcome treatment resistance and enrich the therapeutic arsenal in the new era of personalized medicine. An overview of EOC multifaceted heterogeneity and the key hallmarks of drug-resistance (Figure 3) may enable successful clinical translation into effective targeting therapeutics towards the prioritization of personalized care for resistant disease.

## Figures and Tables

**Figure 1 cells-13-00611-f001:**
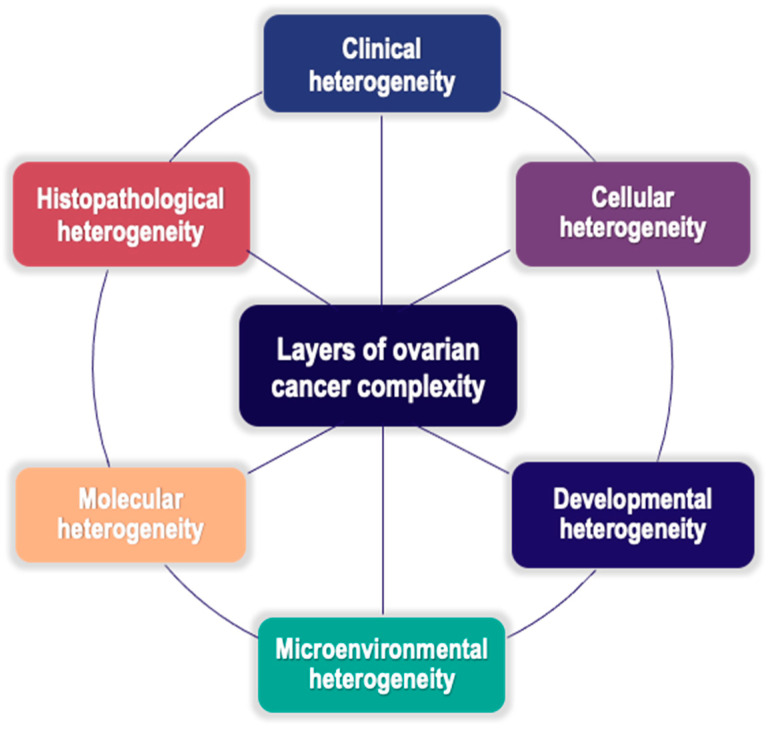
Factors implicated in EOC stem cell heterogeneity and the complex relationships between them. The overall heterogeneity of EOC is deemed to be the outcome of several interconnected mechanisms of diversification, mostly involving the subpopulation of CSCs both directly and indirectly. These mechanisms establish several layers of tumor complexity acting throughout the entire tumor evolution path. Adapted from J. Hatina et al. Ovarian Cancer Stem Cell Heterogeneity. In A. Birbrair (Ed.), *Stem Cells Heterogeneity in Cancer* (pp. 201–216) [8].

**Figure 2 cells-13-00611-f002:**
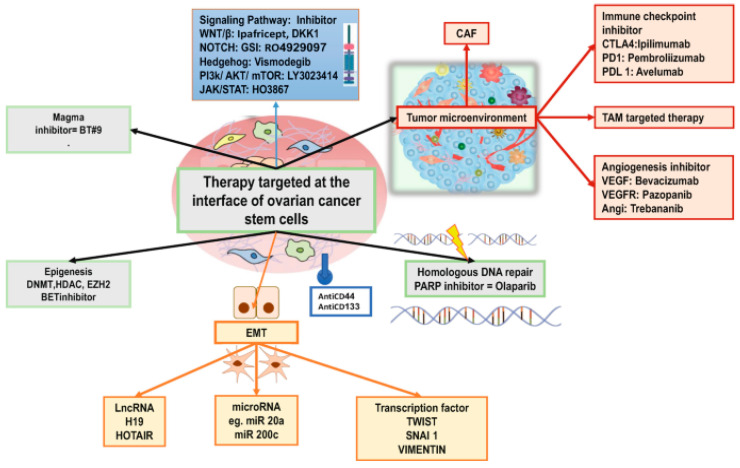
Potential targeting approaches to eradicate drug-resistant CSCs in EOC. Potential targets for the elimination of ovarian cancer stem cells. CAF: cancer-associated fibroblast, VEGF: vascular endothelial growth factor, VEGFR: VEGF receptor, EMT: epithelial to mesenchymal transition, Magma: mitochondrial associated granulocyte macrophage colony stimulating factor, LncRNA: long noncoding RNA, EZH2: enhancer of zeste homolog 2, TAM: tumor-associated macrophage, PD1: programmed death 1, PDL1: programmed death-1 (PD-1) ligand1, PARP: poly (ADP-ribose) polymerase 1, DKK1: Dickopf 1, GSI: gamma secretase inhibitor. Adapted from Saha, S., Parte, S., Roy, P., and Kakar, S. S. (2021). Ovarian cancer stem cells: Characterization and role in tumorigenesis. *Advances in Experimental Medicine and Biology*, 1330, 151–169 [24,34].

**Figure 3 cells-13-00611-f003:**
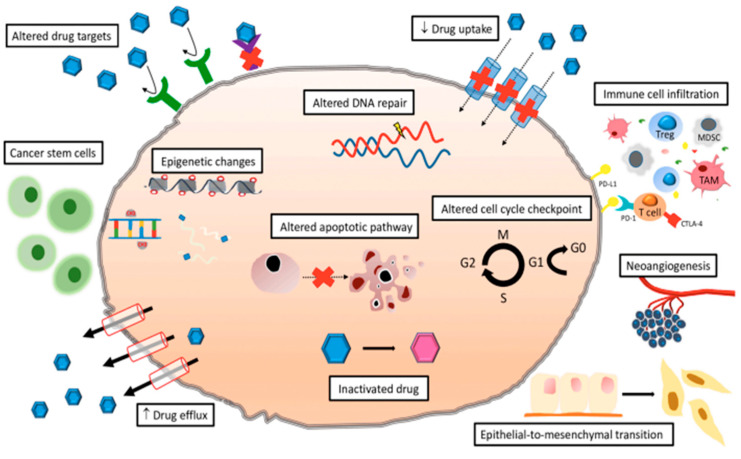
Overview of mechanisms of drug-resistance in EOC. Abbreviations. CTLA-4: cytotoxic T lymphocyte-associated antigen 4. DNA: deoxyribonucleic acid. MDSC: myeloid-derived suppressor cell. PD-1: programmed cell death protein 1. PD-L1: programmed cell death protein ligand 1. TAM: tumor-associated macrophage. Treg: regulatory T cell. Adapted from Marchetti C. et al. Chemotherapy resistance in epithelial ovarian cancer: Mechanisms and emerging treatments. *Seminars in Cancer Biology* 77 (2021) 144–166 [6].

## Data Availability

Data are contained within the article and Appendix A.

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
