# Peer review of "Spotlight on New Hallmarks of Drug-Resistance towards Personalized Care for Epithelial Ovarian Cancer"

_cells, 2024, doi:10.3390/cells13070611_

Round 1

Reviewer 1 Report

Comments and Suggestions for Authors

This is an informative review written by the authors. While the authors claim that there are no biomarkers of chemoresistance, could you please touch on the use of CA125 after the first cycle of chemotherapy on the survival of patients with advanced ovarian cancer (PMID: 26996555)?

In line 135, the authors mentioned that “EOC is a stem cell (SC)-driven tumor type”. Could you please provide citation and the extent to which SC drives EOC? Later, the authors say the exact cell origin of EOC is elusive. The paragraph in section #3.1.1 was conceptually confusing.

When mentioning platinum resistance, please indicate whether it is based on the finding using carboplatin or cisplatin treatment? 

Suggestion: While it was written in the text, it would be clearer to have a subheading on clinical and histopathological heterogeneity.

Please provide citations for the following:

  1. Line 250
  2. Line 252
  3. Line 277-279

The only other concern is the lack of key citations in the field.

Comments on the Quality of English Language

Only a minor editing in English is required. 

Example:

Lines 125-127 are unclear

Reword: Line 360-361.

Author Response

Dear reviewer, thank you for taking the time to review this manuscript. Please find the detailed responses in the attached file below and the corresponding revisions highlighted in the re-submitted files.

Reviewer 2 Report

Comments and Suggestions for Authors

Frezzini and Lonardi wrote a review on chemoresistance in epithelial ovarian cancer. Their work highlighted some of the new areas being studied for chemoresistance, focusing primarily on cancer stem cells, the tumor microenvironment, and molecular heterogeneity. This is a nice general overview of chemoresistance research to date, while not delving into any areas in complex detail. In addition, treatment options as well as biomarkers and liquid biopsies are discussed for tracking and treating chemoresistance. 

A few minor comments to address:

-            The statement after the Results heading is unclear. Please clarify what you mean here as it almost appears as directions for writing the review that were left in.

-            Line 104 mentions that 90% of EOCs arise from the ovarian surface epithelium, but that is not thought to be the case for HGSOCs which is discussed later 114/115 (and section 3.1.1) as arising from the STICs which occur in the fallopian tube. Please clarify this.

-            Minor word choices such as metastasization line 167, next 414, etc

-            Formatting for liquid biopsies seems out of context (line 373) from the rest of the document

-            It might be helpful to have a figure at the end of the review to highlight some of the areas leading to chemoresistance and how some of the treatments target those areas (ie CSC, TME, exosomes, miRNA, signaling etc) to sum up the review. 

-            Supplementary files: Make sure that they are easy to follow. As an example, S4 is hard to follow as to what treatment is in what phase trial and the outcome. Can it be reformatted so that all of the information related to each trial is on one line so that it’s much easier to follow? 

Author Response

(The authors gave the same response as above.)
